

# Phylogenetic comparison between Type IX Secretion System (T9SS) protein components suggests evidence of horizontal gene transfer

Reeki Emrizal and Nor Azlan Nor Muhammad

Institute of Systems Biology, Universiti Kebangsaan Malaysia, Bangi, Selangor, Malaysia

## ABSTRACT

*Porphyromonas gingivalis* is one of the major bacteria that causes periodontitis. Chronic periodontitis is a severe form of periodontal disease that ultimately leads to tooth loss. Virulence factors that contribute to periodontitis are secreted by Type IX Secretion System (T9SS). There are aspects of T9SS protein components that have yet to be characterised. Thus, the aim of this study is to investigate the phylogenetic relationship between members of 20 T9SS component protein families. The Bayesian Inference (BI) trees for 19 T9SS protein components exhibit monophyletic clades for all major classes under Bacteroidetes with strong support for the monophyletic clades or its subclades that is consistent with phylogeny exhibited by the constructed BI tree of 16S rRNA. The BI tree of PorR is different from the 19 BI trees of T9SS protein components as it does not exhibit monophyletic clades for all major classes under Bacteroidetes. There is strong support for the phylogeny exhibited by the BI tree of PorR which deviates from the phylogeny based on 16S rRNA. Hence, it is possible that the *porR* gene is subjected to horizontal transfer as it is known that virulence factor genes could be horizontally transferred. Seven genes (*porR* included) that are involved in the biosynthesis of A-LPS are found to be flanked by insertion sequences (IS5 family transposons). Therefore, the intervening DNA segment that contains the *porR* gene might be transposed and subjected to conjugative transfer. Thus, the seven genes can be co-transferred via horizontal gene transfer. The BI tree of UgdA does not exhibit monophyletic clades for all major classes under Bacteroidetes which is similar to the BI tree of PorR (both are a part of the seven genes). Both BI trees also exhibit similar topology as the four identified clusters with strong support and have similar relative positions to each other in both BI trees. This reinforces the possibility that *porR* and the other six genes might be horizontally transferred. Other than the BI tree of PorR, the 19 other BI trees of T9SS protein components also exhibit evidence of horizontal gene transfer. However, their genes might undergo horizontal gene transfer less frequently compared to *porR* because the intervening DNA segment that contains *porR* is easily exchanged between bacteria under Bacteroidetes due to the presence of insertion sequences (IS5 family transposons) that flank it. In conclusion, this study can provide a better understanding about the phylogeny of T9SS protein components.

Corresponding author
Nor Azlan Nor Muhammad,
norazlannm@ukm.edu.my

## INTRODUCTION

Periodontitis is a form of periodontal disease that is driven by the inflammatory conditions that have deteriorating effects on the structures that support the teeth, including gingiva (gum), alveolar bone, and periodontal ligament. Prolonged inflammatory conditions in chronic periodontitis can cause the destruction of those supporting structures that ultimately leads to tooth loss and might contribute to systemic inflammation (*Kinane, Stathopoulou & Papapanou, 2017*; *Escobar et al., 2018*). This is evidenced by its implications in systemic diseases such as atherosclerosis (*Gotsman et al., 2007*), aspiration pneumonia (*Benedyk et al., 2016*), cancer (*Gao et al., 2016*), rheumatoid arthritis (*Laugisch et al., 2016*), and diabetes mellitus (*Khader et al., 2006*). *Porphyromonas gingivalis* is an oral pathogen that is frequently associated with periodontitis and it is found to acquire Type IX Secretion System (T9SS); a bacterial secretion system that is unique to gram-negative bacteria under the Bacteroidetes phylum (*Sato et al., 2010*).

T9SS exhibits diverse roles among species of bacteria under Bacteroidetes. Other than transporting virulence factors such as gingipains and peptidylarginine deiminase in *P. gingivalis* that can cause human oral diseases (*Potempa, Pike & Travis, 1995*; *Maresz et al., 2013*), T9SS also transports virulence factors such as chondroitin sulfate lyases that can cause columnaris disease which is a form of fish disease. *Flavobacterium columnare*, a fish pathogen that contributes to the epidemic that occurred among wild and cultured fish, is found to acquire T9SS. This epidemic poses a problem to the aquaculture industry as columnaris disease can significantly increase the mortality rate among cultured fish, thus threatening the industry output (*Li et al., 2017*). T9SS is also involved in the transport of non-virulence factors such as cargo proteins that form the bacterial gliding motility apparatus in *Flavobacterium johnsoniae* that aids in its motility (*Nakane et al., 2013*) and enzymes that are important for lignocellulose digestion in the rumen of ruminants that become the hosts for *Candidatus Paraporphyromonas polyenzymogenes* (*Naas et al., 2018*).

Gram-negative bacteria have an outer membrane (OM) that acts as an impermeable layer that prevents the free movement of hydrophilic and hydrophobic molecules across it. This is because of the presence of lipopolysaccharides (LPS) within the outer leaflet of the OM. Outer membrane proteins that are embedded in the OM usually form a channel to allow small molecules to pass through it (*Nikaido, 2003*; *Hong et al., 2006*). However, large molecules such as proteins require larger channels to pass through the OM. Hence, secretion systems are developed by bacteria to enable coordinated transport of specific cargo proteins across the OM. Currently, there are nine different types of secretion systems evolved by bacteria. T9SS is restricted to bacteria under Bacteroidetes (*Sato et al., 2010*; *Lasica et al., 2017*).

T9SS consists of many different protein components that perform coordinated roles to ensure proper translocation and modification of its cargo proteins. These roles can be

categorised into four major functions: translocation, modification, energetic, and regulation (*Sato et al., 2010*; *Lasica et al., 2017*; *Naito et al., 2019*). Initially, the cargo proteins of T9SS are translocated across the inner membrane (IM) via Sec translocon where the signal peptide (SP) of cargo proteins is cleaved (*Rahman et al., 2003*). The cargo proteins also acquire a C-terminal domain (CTD) that interacts with the $PorK_2L_3M_2N_2$ trans-envelope complex to translocate cargo proteins across the periplasm (*Vincent et al., 2017*; *Vincent, Chabalier & Cascales, 2018*) (Fig. 1). PorE has been suggested to form the scaffold of the periplasm complex that translocates cargo proteins across the periplasm (*Heath et al., 2016*; *Naito et al., 2019*). SprA (ortholog of Sov in *F. johnsoniae*) has been proposed as the secretion pore that translocates cargo proteins across the OM (*Lauber et al., 2018*). PorV acts as an outer membrane shuttle protein that delivers the cargo proteins to the attachment complex (*Glew et al., 2017*) (Fig. 1). In the attachment complex, PorU cleaves the CTD of cargo protein. Then, it is glycosylated with anionic lipopolysaccharide (A-LPS) delivered by PorZ at the cleaved site (*Glew et al., 2012*; *Glew et al., 2017*). After both post-translational modifications, the cargo protein will be anchored to the cell surface by A-LPS (*Lasica et al., 2016*; *Glew et al., 2017*) (Fig. 1). PorX and PorY forms a two-component system (TCS) that regulates the operon of *por* genes (*porP*, *porK*, *porL*, *porM*, and *porN*) via SigP (*Vincent et al., 2017*; *Kadowaki et al., 2016*) (Fig. 1). PorR is an aminotransferase that is involved in the Wbp pathway that biosynthesises the structural repeating unit of anionic polysaccharide (APS) (*Shoji et al., 2002*; *Shoji et al., 2014*) (Fig. 1). Despite that, there are T9SS components without known functions (PorP, PorT, PorW, Omp17, PorF, and PorG) (Fig. 1) and a few aspects of T9SS components that have yet to be characterised (*Nguyen et al., 2009*; *Saiki & Konishi, 2010*; *Sato et al., 2010*; *Gorasia et al., 2016*; *Naito et al., 2019*; *Taguchi et al., 2016*).

This work aims to characterise the phylogeny of T9SS protein components. Phylogenetic analysis was performed on the members of 20 T9SS component protein families that have been reported (*Emrizal & Muhammad, 2018*). The Bayesian Inference (BI) trees for 19 T9SS protein components exhibit monophyletic clades for all major classes under Bacteroidetes with strong support for the monophyletic clades or its subclades that is consistent with phylogeny exhibited by the constructed BI tree of 16S rRNA. The BI tree of PorR is different from the other 19 BI trees as it does not exhibit monophyletic clades for all major classes under Bacteroidetes. There is also strong support for the phylogeny exhibited by the BI tree of PorR. Thus, there is a possibility that the *porR* gene is subjected to horizontal transfer as it is known that virulence factor genes could be horizontally transferred (*Hirt, Schlievert & Dunny, 2002*). Seven genes including *porR* that are involved in the biosynthesis of A-LPS are found to be flanked by insertion sequences (IS5 family transposons). This suggests that the intervening DNA segment that contains *porR* can be transposed and subjected to conjugative transfer (*Thomas & Nielsen, 2005*; *Brochet et al., 2009*). Thus, the seven genes might be co-transferred via horizontal gene transfer. The BI trees of PorR and UgdA (both are a part of the seven genes) exhibit similarities. This reinforces the possibility that *porR* and the other six genes might undergo horizontal gene transfer. Other than the BI tree of PorR, the BI trees of the other 19 components also exhibit evidence of horizontal gene transfer. However, for the genes that encode those 19 components, they might undergo horizontal gene transfer less frequently compared to *porR* because the intervening DNA
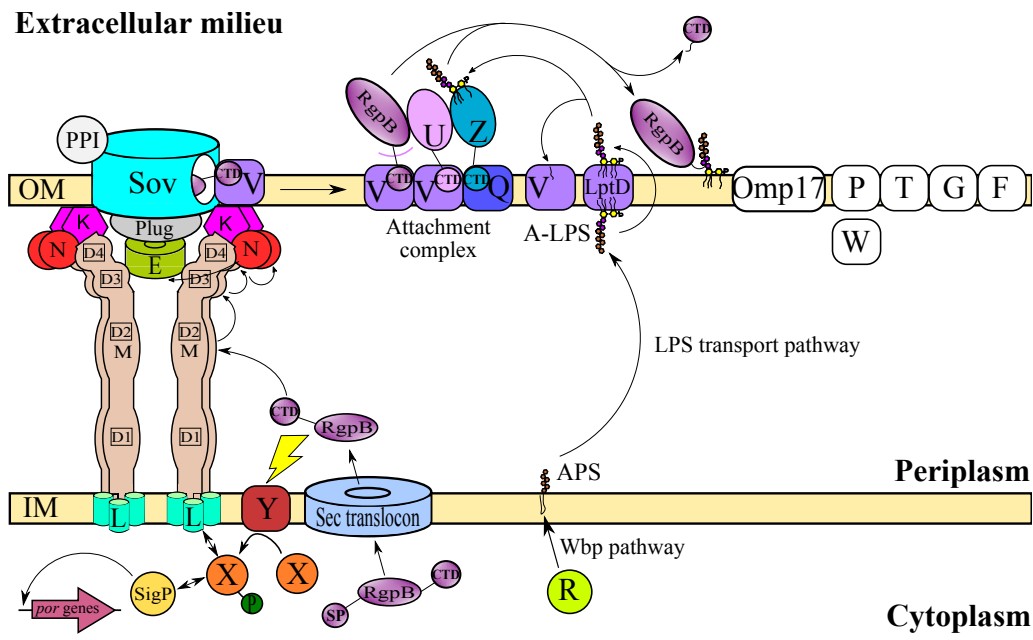

**Figure 1** **T9SS protein components on the inner membrane (IM) and outer membrane (OM) of *Porphyromonas gingivalis*.** The protein components with known functions are represented by coloured structures. The pathway for cargo protein gingipain (RgpB) translocation and modifications by T9SS is illustrated. The regulation of the pathway by the protein components is also exhibited.

segment that contains *porR* is easily exchanged between bacteria under Bacteroidetes due to the presence of IS5 family transposons that flank it.

## MATERIALS & METHODS

### Construction of multiple sequence alignments of T9SS protein components

The multiple sequence alignments for each T9SS protein component were built using the putative members of T9SS component protein families. The pipeline that was used to select those members has been reported (*Emrizal & Muhammad, 2018*). The pipeline was used to filter out false positives among the homologs that have been identified through homology searching using BLASTP which was performed using T9SS component protein sequences retrieved from the NCBI protein database that were searched against a local BLAST database constructed from completely sequenced bacterial proteomes from GenBank. The selection criteria used in the pipeline (e-value ≤ 0.001, query coverage >60%, and Bacteroidetes homolog with the lowest e-value for bacterial strains with multiple hits) can minimise the possibility of false positive inclusion (*Emrizal & Muhammad, 2018*). The sequences of protein homologs used to build the multiple sequence alignments for each T9SS component were provided in FASTA format as (Data S1).

The multiple sequence alignments were constructed using MAFFT (version 7.402) (*Katoh et al., 2002*) on the CIPRES computing cluster (*Miller, Pfeiffer & Schwartz, 2010*) in FASTA format. Unreliable alignment regions in the multiple sequence alignments were

assessed using GUIDANCE2 (version 2.02) (*Sela et al., 2015*) on the CIPRES computing cluster (*Miller, Pfeiffer & Schwartz, 2010*). Columns with low confidence were removed from the multiple sequence alignments. The format of multiple sequence alignments was converted into relaxed interleaved PHYLIP format using an online Format Converter (https://www.hiv.lanl.gov/content/sequence/FORMAT_CONVERSION/form.html). The multiple sequence alignments in relaxed interleaved PHYLIP format were manually edited into NEXUS format.

### Determination of amino acid substitution models for multiple sequence alignments of T9SS protein components

The multiple sequence alignments in relaxed interleaved PHYLIP format (Data S2) were used by ProtTest (version 3.4.2) (*Guindon & Gascuel, 2003*; *Darriba et al., 2011*) to determine the amino acid substitution model to be used for each alignment in the phylogenetic analysis. The graphical user interface (GUI) version of ProtTest was used to test each alignment against 10 amino acid substitution model matrices (Blosum62, CpREV, Dayhoff, JTT, MtMam, MtREV, RtREV, VT, WAG, and LG) with any combination of among-site rate variation (no rate variation across sites, gamma-shaped rate variation across sites (+G), a proportion of invariable sites (+I), or gamma-shaped rate variation across sites with a proportion of invariable sites (+G+I)) and stationary amino acid frequencies (Dirichlet or fixed (empirical) (+F)). The best model according to Bayesian Information Criterion (BIC) (*Schwarz, 1978*) was selected to be used in the phylogenetic analysis for that alignment.

### Bayesian Inference (BI) analysis for multiple sequence alignments of T9SS protein components

Bayesian Inference (BI) analysis was performed using multiple sequence alignments in NEXUS format (Data S3). The BI analysis was performed using MrBayes (version 3.2.6) (*Huelsenbeck & Ronquist, 2001*) on the CIPRES computing cluster (*Miller, Pfeiffer & Schwartz, 2010*) for alignments of 14 components (PorK, PorL, PorM, PorN, PorP, PorQ, PorT, PorU, PorV, SigP, Omp17, PorE, PorF, and PorG). The BI analysis for each alignment was performed with the selected amino acid substitution model and two independent runs for 50,000,000 generations, each with four chains, with a sampling frequency of every 5,000, and a burn-in of 25%. Beagle CPU was utilised to speed up the BI analysis.

The BI analysis for the other 6 components (PorR, Sov, PorW, PorX, PorY, and PorZ) was performed using command-line MrBayes (version 3.2.6) (*Huelsenbeck & Ronquist, 2001*) on a desktop with Nvidia Titan V GPU and CUDA driver (version 10.1) installed. The BI analysis for each alignment was performed with the selected amino acid substitution model and two independent runs for 50,000,000 generations, each with four chains (PorR, Sov, PorW) or eight chains (PorX, PorY, and PorZ), with a sampling frequency of every 5,000, and a burn-in of 25%. Beagle GPU was utilised to speed up the BI analysis. The constructed BI trees were visualised and annotated using online iTOL (version 4.4.2) (*Letunic & Bork, 2019*).

## Construction of Bayesian Inference (BI) tree of 16S ribosomal RNA (rRNA)

The 16S ribosomal RNA (rRNA) sequences have been used to construct the current universal tree of life (*Winker & Woese, 1991*; *Pylro et al., 2012*). Thus, the BI tree of 16S rRNA has been constructed in this work to compare it with the BI trees of T9SS protein components. A pre-formatted BLAST database of microbial 16S rRNA sequences was retrieved from NCBI (ftp://ftp.ncbi.nlm.nih.gov/blast/db/). The 16S rRNA sequence from *Porphyromonas gingivalis* ATCC 33277 (NR_040838.1) was retrieved from NCBI (https://www.ncbi.nlm.nih.gov/gene) and it was searched against that database using local BLASTN (*Altschul et al., 1990*). The pipeline mentioned above was used to select homologs of 16S rRNA gene in species under Bacteroidetes that were also found to acquire homologs of T9SS protein components in this work. For those species that their 16S rRNA sequences could not be retrieved from the microbial 16S rRNA BLAST database, their 16S rRNA sequences were retrieved directly from either NCBI Gene (https://www.ncbi.nlm.nih.gov/gene) or NCBI Nucleotide (https://www.ncbi.nlm.nih.gov/ nuccore/). The selected 16S rRNA sequences were provided in FASTA format as a (Data S1).

The sequences were used to build the multiple sequence alignment of 16S rRNA using MAFFT (version 7.402) (*Katoh et al., 2002*) and unreliable alignment regions in the multiple sequence alignment were assessed using GUIDANCE2 (version 2.02) (*Sela et al., 2015*) on the CIPRES computing cluster (*Miller, Pfeiffer & Schwartz, 2010*). Columns with low confidence were removed from the multiple sequence alignment. The alignment in FASTA format (Data S2) was used to determine the best nucleotide substitution model to be used in the phylogenetic analysis. The graphical user interface (GUI) version of ModelTest (*Darriba et al., 2020*) was used to test the alignment against three nucleotide substitution model matrices (GTR, HKY85, and F81) with any combination of among-site rate variation (no rate variation across sites, gamma-shaped rate variation across sites (+G), a proportion of invariable sites (+I), or gamma-shaped rate variation across sites with a proportion of invariable sites (+G+I)) and stationary amino acid frequencies (Dirichlet or fixed (empirical) (+F)). The best model according to Bayesian Information Criterion (BIC) (*Schwarz, 1978*) was selected to be used in the phylogenetic analysis for that alignment.

BI analysis was performed using the alignment in NEXUS format (Data S3). The analysis was performed using MrBayes (version 3.2.6) (*Huelsenbeck & Ronquist, 2001*) on the CIPRES computing cluster (*Miller, Pfeiffer & Schwartz, 2010*) with the selected nucleotide substitution model and two independent runs for 50,000,000 generations, each with four chains, with a sampling frequency of every 5,000, and a burn-in of 25%. Beagle CPU was utilised to speed up the BI analysis. The BI tree of 16S rRNA was visualised and annotated using online iTOL (version 4.4.2) (*Letunic & Bork, 2019*).

## Identification of *porR* and its neighbouring genes' arrangement in *Porphyromonas gingivalis* ATCC 33277 genome

The sequence of *P. gingivalis* ATCC 33277 genome and annotation files of the genome were retrieved from Genbank (*Naito et al., 2008*). The *P. gingivalis* ATCC 33277 genome

sequence and its annotation files were provided in the (Data S4). The part of *P. gingivalis* ATCC 33277 genome sequence that contains the *porR* and its neighbouring genes was extracted. Then, it was searched against the non-redundant protein sequences (nr) database using online BLASTX. The search was narrowed down to the proteome of *P. gingivalis* ATCC 33277 only. The maximum target sequences were set at the highest value available which is 20,000. Other parameters were left at its default values (*Altschul et al., 1990*). Only the matches with 100% percentage identity and 0 e-value were used to annotate the part of *P. gingivalis* ATCC 33277 genome sequence that contains the *porR* gene.

## Construction of Bayesian Inference (BI) tree of UgdA

Based on the identification of *porR* neighbouring genes, the two genes that are involved in the Wbp pathway (*ugdA* and *porR*) are found to be within the intervening DNA segment that is flanked by IS5 family transposons. Thus, the BI tree of UgdA was constructed to be compared with the BI tree of PorR. The pipeline mentioned above was used to select homologs of UgdA (Data S1) and construct the multiple sequence alignment of UgdA with low confidence columns being removed. The alignment in relaxed interleaved PHYLIP format (Data S2) was used to determine the best amino acid substitution model. BI analysis was performed using UgdA alignment in NEXUS format (Data S3). The analysis was performed using command-line MrBayes (version 3.2.6) (*Huelsenbeck & Ronquist, 2001*) on a desktop with Nvidia Titan V GPU and CUDA driver (version 10.1) installed with the selected amino acid substitution model and two independent runs for 50,000,000 generations, each with four chains, with a sampling frequency of every 5,000, and a burn-in of 25%. Beagle GPU was utilised to speed up the BI analysis. The constructed BI tree was visualised and annotated using online iTOL (version 4.4.2) (*Letunic & Bork, 2019*).

## RESULTS

### Bayesian Inference (BI) trees of T9SS protein components

Bayesian Inference (BI) trees are constructed from the multiple sequence alignments of putative members of T9SS component protein families that have been reported (*Emrizal & Muhammad, 2018*). The characteristics of alignments and the best amino acid substitution model that has been selected for each alignment are shown in Table 1. The selected amino acid substitution model for each alignment defines the parameters that were used for BI analysis for each alignment. The unrooted BI trees of T9SS protein components are shown (Figs. 2–6). The identified monophyletic clades that were formed by terminal nodes that belong to the same class under Bacteroidetes are denoted by solid curves (Figs. 2–6). The monophyletic clades or its subclades with strong support (posterior probability value > 0.95) are denoted by dashed curves (Figs. 2–6).

Out of 20 BI trees of T9SS protein components, only 19 exhibit monophyletic clades for all major classes under Bacteroidetes (Figs. 2–6). Major classes are those with more than five families under the class (Bacteroidia, Cytophagia, and Flavobacteriia) while minor classes are those with less than or equal to five families under the class (Chitinophagia, Sphingobacteriia, Saprospiria, Incertae sedis, and unclassified). Nine of the BI trees (PorK, Sov, PorT, PorV, PorW, PorX, Omp17, PorE, and PorF) exhibit monophyletic clades

**Table 1 The characteristics of T9SS component protein alignments and the best amino acid substitution models that have been selected for them.** The characteristics of T9SS component protein alignments such as number of taxa used to construct the alignments and umber of characters of the alignments are provided. The best amino acid substitution model that has been selected for each alignment is also provided. The definition of parameters of the best amino acid substitution models are provided in the footnote.

| Alignment | No. of taxa | No. of characters | Model |
|---|---|---|---|
| Omp17 | 180 | 245 | LG + G + F |
| PorE | 137 | 793 | LG + G + I |
| PorF | 121 | 829 | LG + G + I + F |
| PorG | 55 | 487 | LG + G + I |
| PorK | 153 | 561 | LG + G + I |
| PorL | 123 | 281 | LG + G + I |
| PorM | 159 | 406 | LG + G + I + F |
| PorN | 62 | 267 | LG + G + I |
| PorP | 138 | 281 | LG + G + I + F |
| PorQ | 108 | 358 | LG + G + F |
| PorR | 176 | 471 | LG + G + I |
| PorT | 151 | 202 | LG + G + F |
| PorU | 109 | 919 | LG + G + I |
| PorV | 162 | 360 | LG + G + I + F |
| PorW | 137 | 995 | LG + G + I + F |
| PorX | 162 | 624 | LG + G + I |
| PorY | 162 | 897 | LG + G + I |
| PorZ | 102 | 569 | LG + G + I |
| SigP | 177 | 197 | LG + G + I |
| Sov | 159 | 2704 | LG + G + I + F |
| UgdA | 176 | 460 | LG + G + I |
| 16S rRNA | 144 | 1452 | GTR + G + I |

**Notes.**

LG + G: LG substitution model matrix with gamma-shaped rate variation across sites and Dirichlet stationary amino acid frequencies.

LG + G + I: LG substitution model matrix with gamma-shaped rate variation across sites with a proportion of invariable sites and Dirichlet stationary amino acid frequencies.

LG + G + I + F: LG substitution model matrix with gamma-shaped rate variation across sites with a proportion of invariable sites and fixed (empirical) stationary amino acid frequencies.

LG + G + F: LG substitution model matrix with gamma-shaped rate variation across sites and fixed (empirical) stationary amino acid frequencies.

GTR + G + I: General Time Reversible (GTR) substitution model matrix with gamma-shaped rate variation across sites with a proportion of invariable sites and Dirichlet stationary nucleotide frequencies.

for all major classes under Bacteroidetes with strong support. Ten of the BI trees (PorL, PorM, PorN, PorP, PorQ, PorU, PorY, PorZ, SigP, and PorG) exhibit strong support for the monophyletic clades or its subclades for all major classes under Bacteroidetes (Figs. 2–6). Despite the presence of PorR homologs from species under Bacteroidia, Cytophagia, and Flavobacteriia, the BI tree of PorR does not exhibit monophyletic clades for all major classes under Bacteroidetes (Fig. 3C). Thus, the BI tree of PorR is different compared to the other 19 BI trees of T9SS protein components that exhibit monophyletic clades for all major classes under Bacteroidetes.

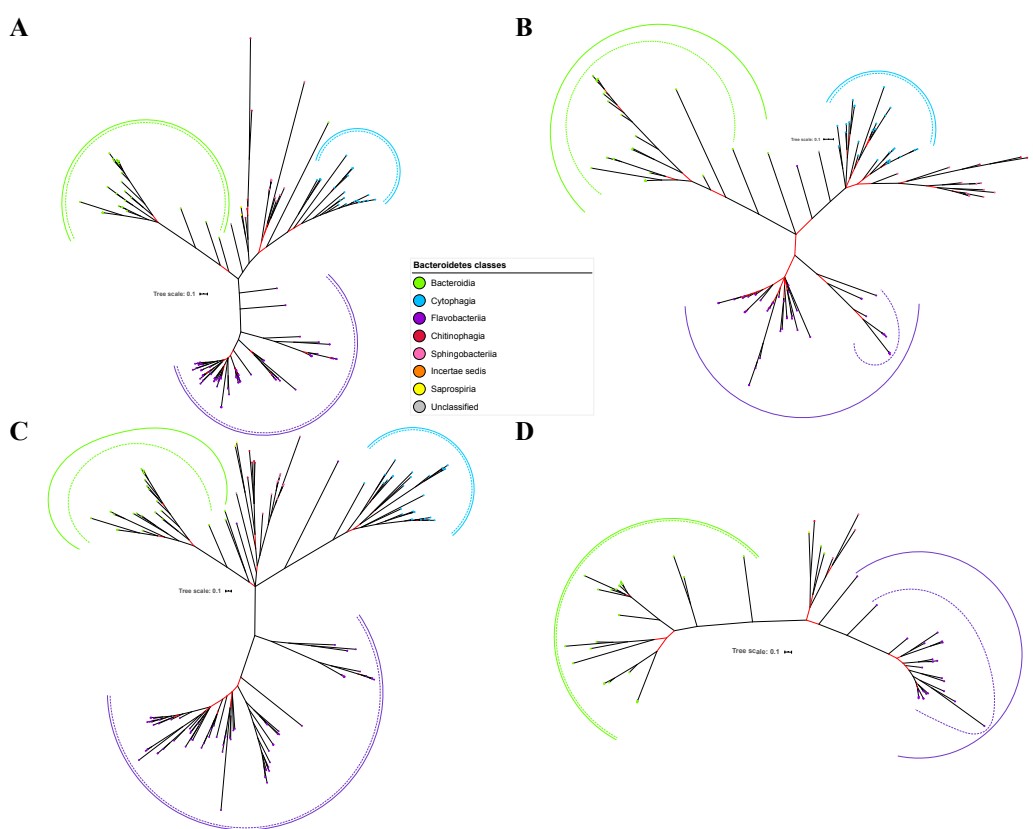

**Figure 2** **The Bayesian Inference (BI) phylogenetic trees of T9SS protein components (PorK, PorL, PorM, and PorN).** (A) BI tree of PorK. (B) BI tree of PorL. (C) BI tree of PorM. (D) BI tree of PorN. The terminal nodes are labelled with coloured circles that represent the classes under Bacteroidetes that each protein homolog belongs to. The classes represented by each colour are provided in the legend inside the figure. The branches with strong support (posterior probability value > 0.95) are coloured in black. Otherwise, the branches are coloured in red. The solid curve denotes a monophyletic clade that was formed by terminal nodes that belong to the same class under Bacteroidetes. The dashed curve denotes a strong support for the monophyletic clade or its subclade. The colour of curve represents the class of terminal nodes that form the clade. The classes represented by each colour are shown in the legend inside the figure.

Some of the terminal nodes of the 19 BI trees of T9SS protein components are out of their expected monophyletic clades (Figs. 2–6). The species corresponding to those terminal nodes are listed in Table S1. There are species that frequently have their terminal nodes out of their expected monophyletic clades such as *Fluviicola taffensis* DSM 16823, bacterium L21-Spi-D4, *Owenweeksia hongkongensis* DSM 17368, and *Draconibacterium orientale*. The terminal nodes corresponding to *F. taffensis* DSM 16823 are found to be out of their expected monophyletic clades in 14 out of 19 BI trees (except PorK, PorN, PorP, PorU, and SigP). The terminal nodes corresponding to bacterium L21-Spi-D4 are found to be out of their monophyletic clades in 10 out of 19 BI trees (except PorK, PorL, PorM, Sov, PorT, PorU, PorX, PorY, and PorE). The terminal nodes corresponding to *O. hongkongensis* DSM 17368 are found to be out of their expected monophyletic clades in 6 out of 19 BI trees (PorM, PorP, PorV, PorY, Omp17, and PorE). The terminal nodes

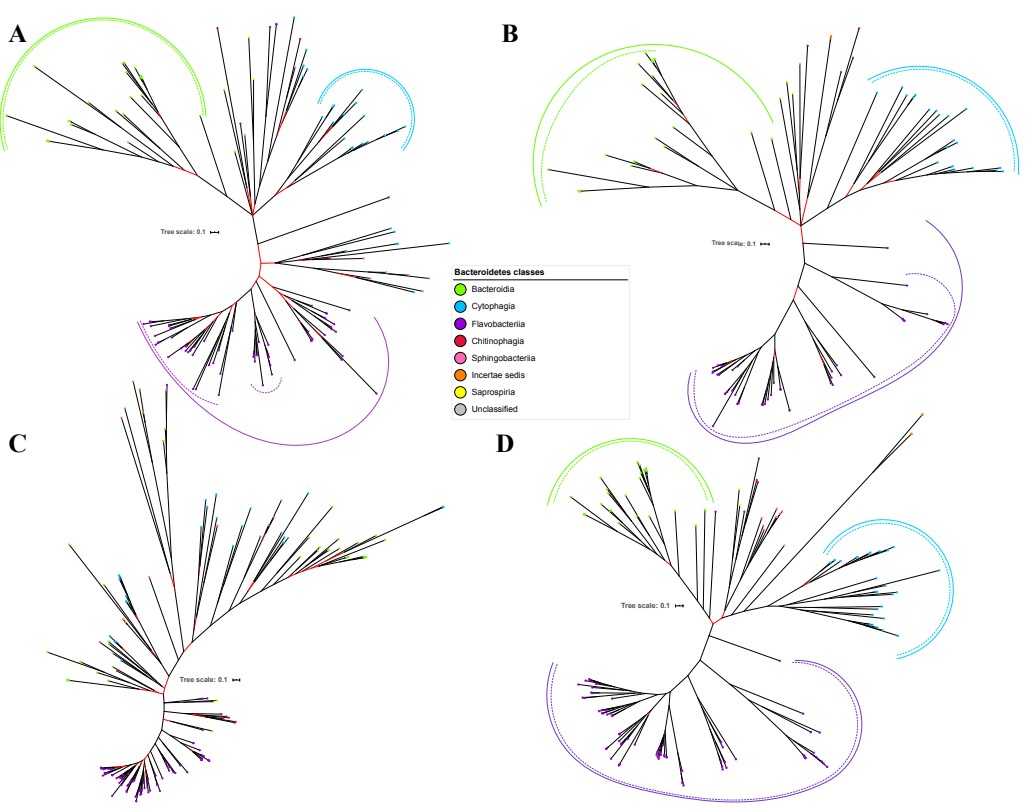

**Figure 3** The Bayesian Inference (BI) phylogenetic trees of T9SS protein components (PorP, PorQ, PorR, and Sov). (A) BI tree of PorP. (B) BI tree of PorQ. (C) BI tree of PorR. (D) BI tree of Sov.

corresponding to *D. orientale* are found to be out of their expected monophyletic clades in 7 out of 19 BI trees (PorN, PorP, PorV, PorW, PorY, SigP, and Omp 17). The 20 BI trees with terminal nodes labelled with their corresponding species and support values for each branch are shown in the (Figs. S1–S20).

## Bayesian Inference (BI) tree of 16S rRNA

The BI tree of 16S rRNA was constructed from the multiple sequence alignment of 16S rRNA homologs from species that were identified to also acquire T9SS protein homologs. Out of 181 species that acquire T9SS protein homologs, only 16S rRNA sequences from 144 species were able to be retrieved from NCBI. The characteristics of 16S rRNA alignment and the best nucleotide substitution model that had been selected for that alignment are shown in Table 1. The unrooted BI tree of 16S rRNA is shown in Fig. 7. The identified monophyletic clades that were formed by terminal nodes that belong to the same class under Bacteroidetes are denoted by solid curves (Fig. 7). The monophyletic clades or its subclades with strong support (posterior probability value >0.95) are denoted by dashed curves (Fig. 7).

The BI tree of 16S rRNA was constructed to be compared to the BI trees of T9SS protein components. The 16S rRNA exhibits monophyletic clades for all major classes

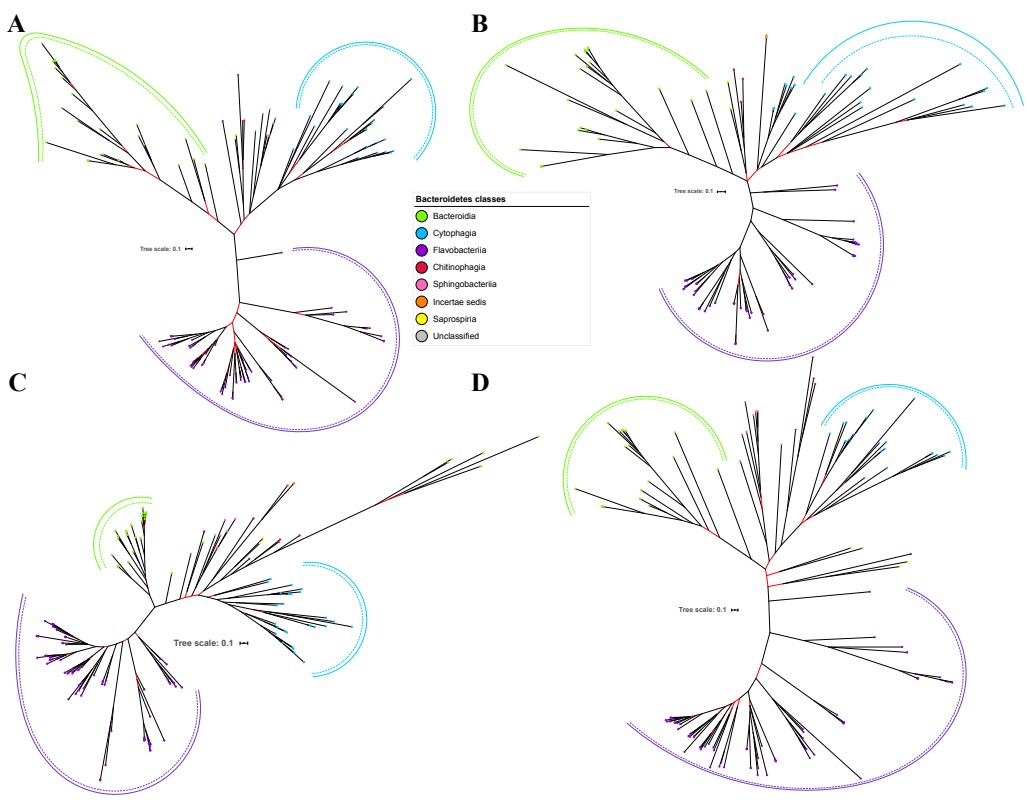

**Figure 4** The Bayesian Inference (BI) phylogenetic trees of T9SS protein components (PorT, PorU, PorV, and PorW). (A) BI tree of PorT. (B) BI tree of PorU. (C) BI tree of PorV. (D) BI tree of PorW.

under Bacteroidetes with strong support (Fig. 7) similar to the 19 BI trees of T9SS protein components. The 16S rRNA also exhibits monophyletic clades for all minor classes under Bacteroidetes with strong support denoted by 4 monophyletic clades of red, pink, yellow, and orange circles (Fig. 7). None of the 20 BI trees of T9SS protein components exhibit phylogeny of the minor classes that is consistent with the phylogeny exhibited by the 16S rRNA tree (Figs. 2–7). Hence, minor classes are excluded in the comparison between 20 BI trees of T9SS protein components. The BI tree of 16S rRNA with terminal nodes labelled with their corresponding species and support values for each branch are shown in the (Fig. S21).

## Arrangement of *porR* and its neighbouring genes in *P. gingivalis* ATCC 33277 genome

As shown in Fig. 8, *porR* and its neighbouring genes are flanked by IS5 family transposons. The IS5 family transposon (cyan rectangles) encodes IS5 family transposase that cleaves the flanking 12 bp inverted repeats (purple triangles) (Fig. 8). This might suggest the possibility that the intervening DNA segment that contains seven genes that are involved in A-LPS biosynthesis (yellow rectangles) can undergo transposition and is possibly subjected to conjugative transfer (Fig. 8) (*Thomas & Nielsen, 2005*; *Brochet et al., 2009*). *porR* (PGN_1236) and *ugdA* (PGN_1243) genes (Fig. 8) have been reported to be involved

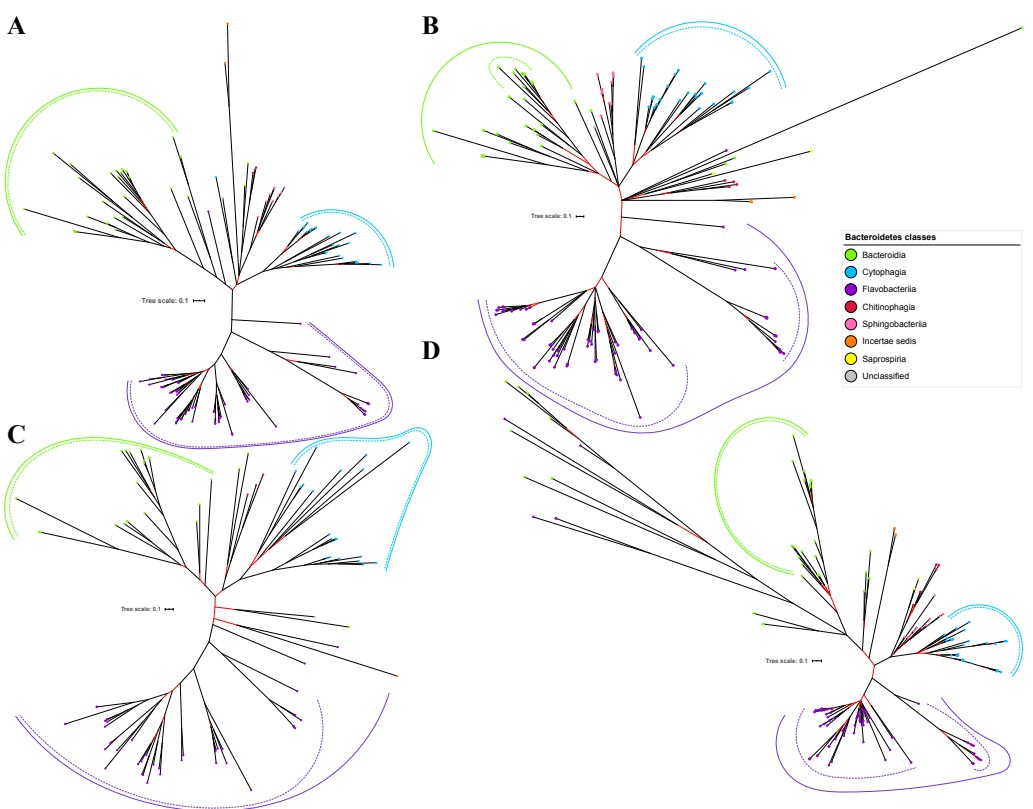

**Figure 5** **The Bayesian Inference (BI) phylogenetic trees of T9SS protein components (PorX, PorY, PorZ, and SigP).** (A) BI tree of PorX. (B) BI tree of PorY. (C) BI tree of PorZ. (D) BI tree of SigP.

in the Wbp pathway that is important for the biosynthesis of structural sugar (di-acetylated glucuronic acid) of A-LPS (*Shoji et al., 2002*; *Shoji et al., 2014*). *porS* (PGN_1235) and *wzy* (PGN_1242) genes (Fig. 8) have been reported to participate in the assembly of A-LPS in bacterial inner membrane (*Shoji et al., 2013*). *gtfB* (PGN_1251) and *gtfE* (PGN_1240) glycosyltransferase genes (Fig. 8) are important for A-LPS biosynthesis while *rfa* (PGN_1255) glycosyltransferase gene (Fig. 8) is important for the biosynthesis of lipid A-core portion of A-LPS (*Shoji et al., 2018*).

## Bayesian Inference (BI) tree of UgdA

The BI tree of UgdA was constructed from the multiple sequence alignment of UgdA homologs that were identified using the same pipeline that has been reported to select T9SS protein homologs (*Emrizal & Muhammad, 2018*). The characteristics of UgdA alignment and the best amino acid substitution model that had been selected for that alignment are shown in Table 1. The unrooted BI tree of UgdA is shown in Fig. 9B. The unrooted BI tree of PorR is also shown in Fig. 9A to be compared with the BI tree of UgdA. Both BI trees do not exhibit monophyletic clades for all major classes under Bacteroidetes. Both BI trees also exhibit similar topology. Four similar clusters (I, II, III, and IV) were identified between both BI trees. Cluster I consists primarily of terminal nodes from Flavobacteriia

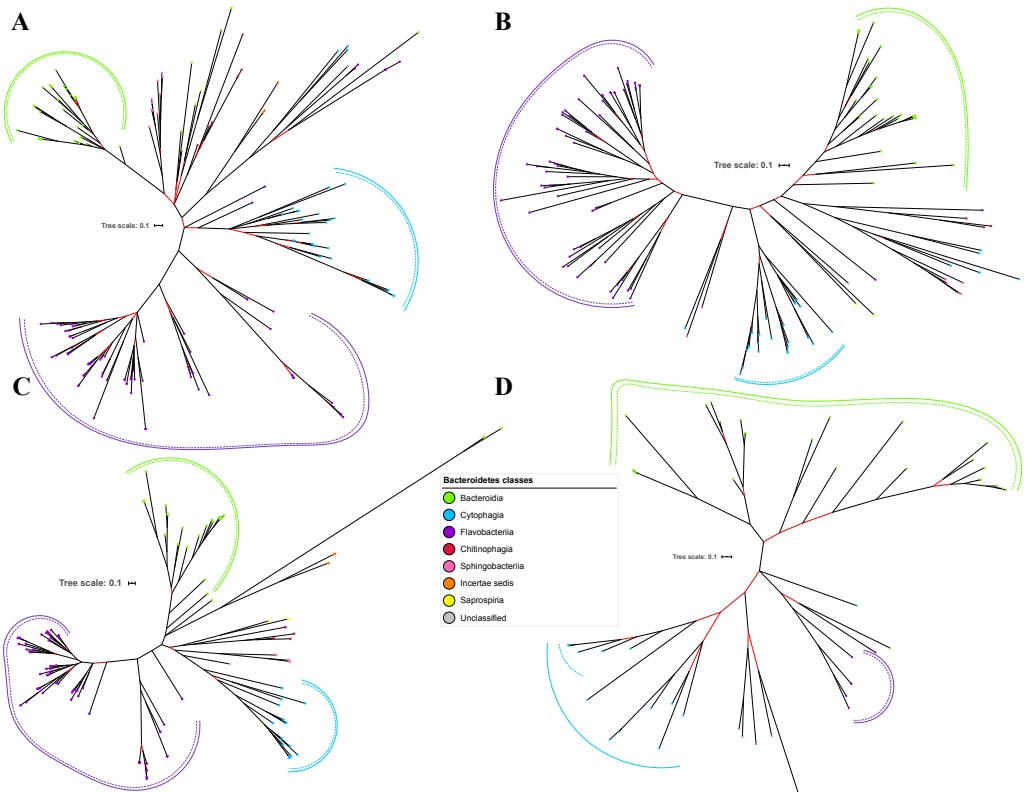

**Figure 6** **The Bayesian Inference (BI) phylogenetic trees of T9SS protein components (Omp17, PorE, PorF, and PorG).** (A) BI tree of Omp17. (B) BI tree of PorE. (C) BI tree of PorF. (D) BI tree of PorG.

and a few terminal nodes from other classes. Cluster II consists of terminal nodes from Porphyromonas, Tannerella, and Parabacteroides genera. Cluster III consists of terminal nodes from Rufibacter and Hymenobacter genera. Cluster IV consists of terminal nodes from Prevotella, Bacteroides, Proteiniphilum, and other genera. The BI tree of UgdA with terminal nodes labelled with their corresponding species and support values for each branch are shown in the (Fig. S22).

## Taxonomic distribution of T9SS protein components

As shown in the 20 BI trees of T9SS components (Figs. 2–6), only bacteria under Bacteroidia, Flavobacteriia, and Chitinophagia classes acquired the 20 components investigated in this work.

The bacteria under Cytophagia class acquired only 19 protein components (except PorN). The bacteria under Saprospiria class acquired only 18 protein components (except PorL and PorG). The bacteria under Sphingobacteriia class acquired only 17 protein components (except PorQ, PorU, and PorZ). The unclassified bacteria acquired only 17 protein components (except PorN, PorU, and PorG). The bacteria under Incertae sedis class acquired only 11 protein components (PorQ, PorR, Sov, PorU, PorV, PorX, PorY, PorZ, SigP, Omp17, and PorF).

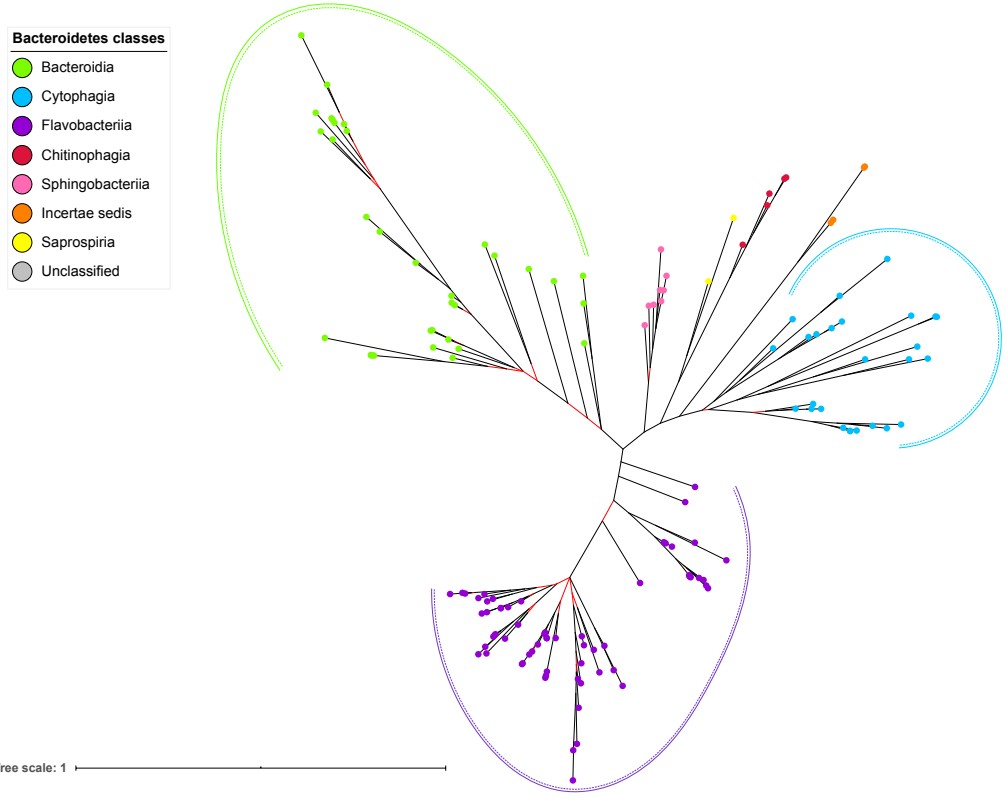

**Figure 7** **The Bayesian Inference (BI) phylogenetic tree of T9SS containing Bacteroidetes species 16S ribosomal RNA (rRNA).** The BI tree of 16S rRNA exhibits monophyletic clades where each clade consists of terminal nodes of the same colour that denotes that they belong to the same class under Bacteroidetes. There is a high support (posterior probability value > 0.95) for each monophyletic clade indicates by the black branch leading to each clade. The solid and dashed green, purple, and blue curves indicate there is a strong support for the monophyletic clades of Bacteroidia, Flavobacteriia, and Cytophagia classes respectively.

The findings in this work are consistent with the taxonomic distribution of T9SS components among bacteria under Bacteroidetes where it has been reported that Bacteroidia, Flavobacteriia, Cytophagia, Sphingobacteriia, and Incertae sedis classes acquired T9SS component homologs (*McBride & Zhu, 2013*). However, comparing the reported taxonomic distribution of T9SS components to the findings in this work, we have identified other species under Chitinophagia, Saprospiria, and those that are unclassified that have acquired T9SS component homologs. Those species and T9SS component homologs they acquired are illustrated in Fig. 10.

## DISCUSSION

The 19 Bayesian Inference (BI) trees of T9SS protein components exhibit monophyletic clades for all major classes under Bacteroidetes with strong support for the monophyletic clades or its subclades (Figs. 2–6). Similar to the 19 BI trees of T9SS protein components, the BI tree of 16S rRNA also exhibits monophyletic clades for all major classes under

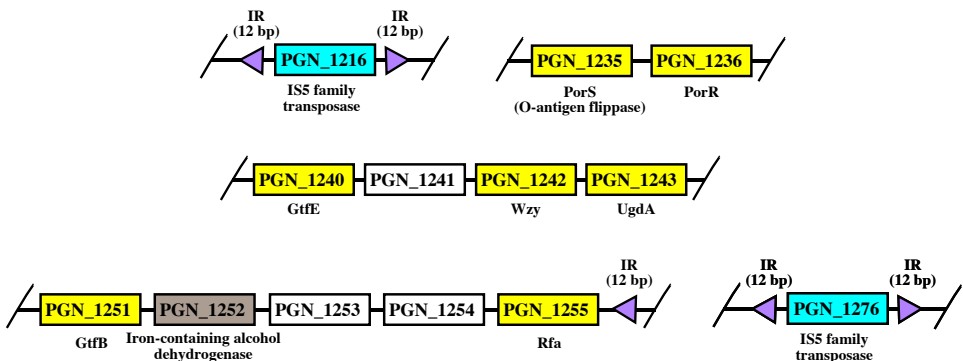

**Figure 8 The arrangement of *porR* and its neighbouring genes in *P. gingivalis* ATCC 33277 genome.** *porR* (PGN_1236) and its neighbouring genes are flanked by IS5 family transposons that formed a composite transposon of 70 kbp in length. The genes that involve in biosynthesis of A-LPS are represented by yellow rectangles while the gene that does not involve is represented by brown rectangle. The genes for hypothetical proteins are represented by white rectangles. The genes for IS5 family transposases are represented by cyan rectangles. The purple triangles represented 12 bp inverted repeats that flanked the genes for IS5 family transposases. Name of proteins encoded by the genes are shown under rectangles that represented the genes. The slashes indicated gaps in the genome.

Bacteroidetes with strong support (Fig. 7). 16S rRNA has been extensively used in phylogenetic analysis for the purpose of evolutionary comparison and classification. The reliability of this approach lies on the assumption that the 16S rRNA gene undergoes hierarchical and unidirectional evolution and no gene transfer of 16S rRNA occurs between species (*Karlsson et al., 2011*). Due to the advantages that the 16S rRNA gene has such as ubiquity in bacterial genomes, being easily sequenced, and widely available in public sequence databases, the current universal tree of life is based on the phylogeny of this gene (*Winker & Woese, 1991*; *Coutinho et al., 1999*; *Pylro et al., 2012*). That assumption has been challenged due to the presence of multiple copies of 16S rRNA in a bacterial genome and the 16S rRNA genes from operons in the same genome are rather distinct which might suggest that such genes might have undergone horizontal gene transfer (*Pei et al., 2010*; *Karlsson et al., 2011*). However, the extent of 16S rRNA evolution remains considerably less compared to the other genes in the bacterial genome (*Espejo & Plaza, 2018*). Thus, 16S rRNA remains relevant for the purpose of evolutionary comparison and classification.

It is expected that the BI trees of T9SS protein components would exhibit similar phylogeny with the BI tree of 16S rRNA. However, the BI trees of T9SS protein components exhibit inconsistent positions of terminal nodes from minor classes among themselves and the phylogeny for the minor classes deviate from the phylogeny exhibited by the BI tree of 16S rRNA (Figs. 2–7). Hence, the minor classes under Bacteroidetes are excluded from the comparison between the 20 BI trees of T9SS protein components. This might arise due to insufficient taxa from minor classes provided to construct those BI trees. Hence, the information that is provided is insufficient to fully resolve the phylogeny of minor classes. As more T9SS-acquiring species from minor classes are sequenced later on, the phylogeny of T9SS protein components will be more resolved (*Alvizu et al., 2018*).

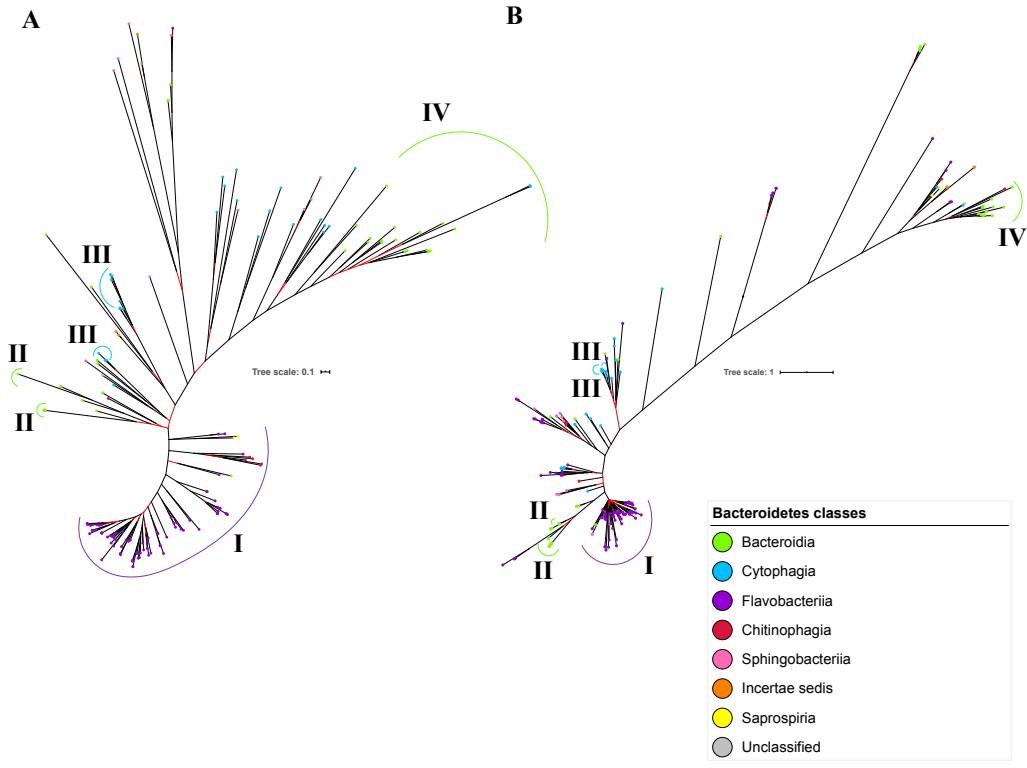

**Figure 9 Comparison between the Bayesian Inference (BI) tree of UgdA with BI tree of PorR.** The BI tree of UgdA (A) does not exhibit monophyletic clades for all major classes under Bacteroidetes which is similar to the BI tree of PorR (B). Both BI trees of UgdA and PorR also exhibit similar topology. Both trees exhibit cluster I (solid purple curve) that primarily consists of terminal nodes of Flavobacteriia and a few terminal nodes from other classes. Both trees have cluster II (solid green curve) that consists of terminal nodes of Porphyromonas, Tannerella, and Parabacteroides genera. Both trees acquire cluster III (solid blue curve) that consists of terminal nodes of Rufibacter and Hymenobacter genera. Both trees exhibit cluster IV (solid green curve) that consists of terminal nodes of Prevotella, Bacteroides, Proteiniphilum, and other genera.

Different from the other 19 BI trees of T9SS protein components, the BI tree of PorR does not exhibit monophyletic clades for all major classes under Bacteroidetes (Figs. 2–6).

The presence of strong support (posterior probability value >0.95) as denoted by the black branch leading to the top half of the BI tree of PorR (Fig. 3C) suggests that there is strong support that the phylogeny exhibited by the BI tree of PorR deviates from the phylogeny based on the 16S rRNA sequence (Fig. 7). Thus, there is a possibility that the *porR* gene is subjected to horizontal transfer hence causing deviation from the expected phylogeny (*Pylro et al., 2012*). Hirt, Schlievert & Dunny have demonstrated that virulence factors and antibiotic resistance genes could be horizontally transferred (*Hirt, Schlievert & Dunny, 2002*). Hence, this suggests the possibility that the *porR* gene that encodes one of the virulence factors produced by *P. gingivalis* can be horizontally transferred (*Shoji et al., 2002*; *Shoji et al., 2014*).

The arrangement of *porR* and its neighbouring genes in the *P. gingivalis* ATCC 33277 genome was identified in order to support the possibility that *porR* is horizontally

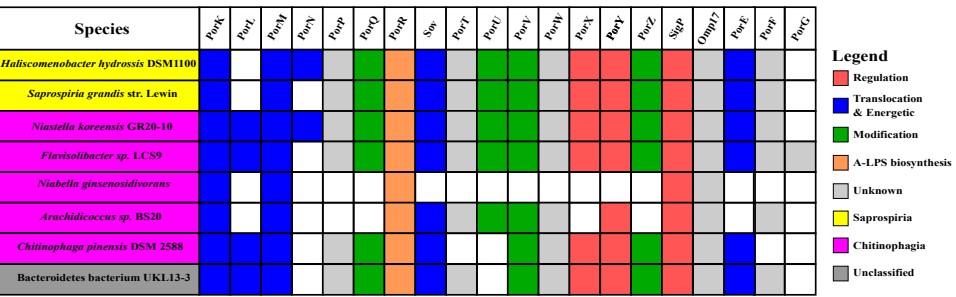

**Figure 10** **The species from Chitinophagia, Saprospiria, and unclassified under Bacteroidetes phylum that acquired homologs of T9SS protein components.** The colours of rectangles denote the classes those species belong to. Coloured squares indicate T9SS component homologs acquired by the species where the different colours denote different functions those components performed. White squares indicate T9SS component homologs absent in those species.

transferred. *P. gingivalis* ATCC 33277 genome was chosen because many gene orthologs that are involved in A-LPS biosynthesis have been identified in this genome (*Shoji et al., 2018*). *porR* and its neighbouring genes are found to be flanked by insertion sequences (IS5 family transposons) (Fig. 8). The IS5 family transposons (cyan rectangles) contain a single open reading frame that encodes for IS5 family transposase that cleaves the 12 bp inverted repeats (purple triangles) that flank the insertion sequences (Fig. 8). The 12 bp inverted repeats show imperfect homology to each other with the consensus sequence: GAGACCTTTG[CA]A. Both of the IS5 family transposons are ∼1300 bp in length. These features are typical of IS5 family transposons (*Mahillon & Chandler, 1998*; *Naito et al., 2008*). The intervening DNA segment and both IS5 family transposons that flank it might form a composite transposon where the cleaving action of IS5 family transposases on inverted repeats can mobilise the intervening DNA segment that contains the *porR* gene and possibly subject it to conjugative transfer (*Thomas & Nielsen, 2005*; *Brochet et al., 2009*). The length of the composite transposon is ∼70 kbp. However, it is also possible for IS5 family transposase to cleave the inverted repeat directly downstream of *rfa* (PGN_1255) (Fig. 8) which will reduce the length of the composite transposon to ∼47 kbp. It has been reported that a transposon of ∼47 kbp in length is able to undergo both transposition and conjugation processes (*Brochet et al., 2009*). Hence, it might be possible for composite transposons of such length to undergo transposition and subsequently be horizontally transferred via bacterial conjugation.

The intervening DNA segment contains seven genes that are involved in the biosynthesis of A-LPS (Fig. 8). Both *porR* (PGN_1236) and *ugdA* (PGN_1243) genes are involved in the Wbp pathway that is important for the biosynthesis of di-acetylated glucuronic acid which is the structural sugar of A-LPS (*Shoji et al., 2002*; *Shoji et al., 2014*). The *porS* gene (PGN_1235), which is an O-antigen flippase, and *wzy* gene (PGN_1242), which is an O-antigen polymerase, are involved in the assembly of A-LPS on the periplasmic side of bacterial IM (*Shoji et al., 2013*). *gtfB* (PGN_1251) and *gtfE* (PGN_1240) glycosyltransferase genes are involved in the biosynthesis of the sugar moiety of A-LPS. *rfa* (PGN_1255)

glycosyltransferase gene is involved in the biosynthesis of the lipid A-core moiety of A-LPS (*Shoji et al., 2018*). However, there are other genes that are involved in the biosynthesis of A-LPS and they are spread out throughout the genome (*Shoji et al., 2018*). Usually, genes that are co-regulated and involved in a similar pathway are clustered in a single operon (*Yanofsky & Lennox, 1959*; *Osbourn & Field, 2009*). Thus, it is possible that the other genes do not form a cluster with the seven genes that are identified to be flanked by insertion sequences (IS5 family transposons) because they are not co-regulated.

It is possible that those seven genes might be co-transferred via horizontal gene transfer. Thus, phylogenetic analysis was performed for the protein alignment of UgdA that is encoded by *ugdA* which, together with *porR*, are involved in the Wbp pathway and are co-localised in the intervening DNA segment flanked by IS5 family transposons (Fig. 8). The BI tree of UgdA (Fig. 9B) was constructed to be compared with the BI tree of PorR (Fig. 9A). Different to the 19 BI trees of T9SS protein components, both BI trees do not exhibit monophyletic clades for all major classes under Bacteroidetes. They also exhibit similar topology where four similar clusters (I, II, III, and IV) with strong support (denoted by a black branch leading to the cluster) have been identified in both BI trees. Cluster I consists of terminal nodes from Flavobacteriia and a few terminal nodes from other classes. Cluster II consists of terminal nodes from Porphyromonas, Tannerella, and Parabacteroides genera. Cluster III consists of terminal nodes from Rufibacter and Hymenobacter genera. Cluster IV consists of terminal nodes from Prevotella, Bacteroides, Proteiniphilum, and other genera. These four clusters exhibit similar relative positions to each other in both BI trees (e.g., cluster I is closer to cluster II than the other clusters and cluster III is closer to cluster II than the other clusters). However, due to the differences in branch lengths between both BI trees, they look slightly different as the upper part of the UgdA tree (Fig. 9B) appears more elongated than the upper part of the PorR tree (Fig. 9A), while the lower part of the UgdA tree (cluster I) appears more shortened than the lower part of the PorR tree (cluster I).

Other than the BI tree of PorR, the BI trees of the other 19 T9SS protein components also exhibit evidence of horizontal gene transfer perhaps between classes under Bacteroidetes. As listed in Table S1, there are terminal nodes that are out of their expected monophyletic clades in the BI trees of those components that suggests the genes that encode them might be horizontally transferred. In theory, the common ancestral species of a monophyletic clade for a class under Bacteroidetes passes the gene that encodes T9SS protein components to its descendant species. Thus, the descendant species that are out of their expected monophyletic clades most likely acquired that gene from the common ancestral species of a monophyletic clade from another class that could be interpreted as a horizontal gene transfer between classes under Bacteroidetes (*Thomas & Nielsen, 2005*; *Brochet et al., 2009*). It is interesting to highlight that there are species that frequently have their corresponding terminal nodes in those 19 BI trees out of their expected monophyletic clades (Figs. 2–6) such as *F. taffensis* DSM 16823, bacterium L21-Spi-D4, *O. hongkongensis* DSM 17368, and *D. orientale*. Thus, it is likely that those bacteria frequently acquire their T9SS components through horizontal gene transfer. However, for the genes that encode those 19 components, they might undergo horizontal gene transfer less frequently compared to *porR* that causes

most of the terminal nodes of BI trees of those components to cluster according to their respective classes. It might be because the intervening DNA segment that contains the *porR* gene is easily exchanged between bacteria under Bacteroidetes due to the presence of insertion sequences (IS5 family transposons) that flank it (Fig. 8).

T9SS is made up of various protein components that form the regulation, translocation, energetic, and modification components. Currently, the secretion system is primarily found in bacteria under the Bacteroidetes phylum (*Abby et al., 2016*). Bacteria from classes under Bacteroidetes (Bacteroidia, Flavobacteriia, Cytophagia, Chitinophagia, Sphingobacteriia, Saprospiria, Incertae sedis, and unclassified) are found to acquire T9SS protein components (Figs. 2–6). However, not all of them acquire the 20 components that have been reported (*Sato et al., 2010*; *Lasica et al., 2017*). As shown in the 20 BI trees of T9SS protein components (Figs. 2–6), only bacteria under Bacteroidia, Flavobacteriia, and Chitinophagia acquired the 20 components investigated. The bacteria under Cytophagia only acquired 19 components (except PorN). The bacteria under Saprospiria only acquired 18 components (except PorL and PorG). The bacteria under Sphingobacteriia only acquired 17 components (except PorQ, PorU, and PorZ). The unclassified bacteria only acquired 17 components (except PorN, PorU, and PorG). The bacteria under Incertae sedis only acquired 11 components (PorQ, PorR, Sov, PorU, PorV, PorX, PorY, PorZ, SigP, Omp17, and PorF). It is interesting to note that PorU, PorZ, and PorQ form the modification components of T9SS. Thus, Sphingobacteriia does not acquire the components that perform post-translational modifications on T9SS cargo proteins such as cleavage of CTD and A-LPS glycosylation. Perhaps, T9SS acquired by Sphingobacteriia does not cleave the CTD of cargo protein and glycosylate it with A-LPS, but leaves the cargo protein bounded to PorV after it is translocated to bacterial cell surface by Sov. Another possible explanation is that Sphingobacteriia does have proteins that perform the functions of missing protein components. However, those proteins exhibit limited sequence similarity with any currently known T9SS protein component. Thus, they could not be detected by the homology searching method used in this work. This explanation could also be applied for other species of bacteria under Bacteroidetes that do not acquire the homologs of the 20 T9SS components.

This work has found other species under Chitinophagia, Saprospiria, and those that are unclassified that acquired homologs of T9SS components that, to our knowledge, might not have been reported (*McBride & Zhu, 2013*). Those other species and the homologs of T9SS components they acquired are shown in Fig. 10. This identification might be due to the analysis that was performed which might cover more bacterial species than previous works as more bacterial genomes have been completely sequenced in the past few years.

## CONCLUSIONS

The objective of this work was to investigate the phylogenetic relationship among putative members of 20 T9SS component protein families (*Emrizal & Muhammad, 2018*). The Bayesian Inference (BI) trees for 19 T9SS protein components exhibit monophyletic clades for all major classes under Bacteroidetes with strong support for the monophyletic clades

or its subclades, which is consistent with the phylogeny exhibited by the constructed BI tree of 16S rRNA. However, the BI tree of PorR is different from the other 19 BI trees of T9SS protein components as it does not exhibit monophyletic clades for all major classes under Bacteroidetes. There is strong support for the phylogeny exhibited by the BI tree of PorR which deviates from the phylogeny based on the 16S rRNA sequence. Thus, there is a possibility that the *porR* gene is subjected to horizontal transfer as it is known that virulence factor genes could be horizontally transferred. Seven genes that are involved in the biosynthesis of A-LPS that includes *porR* are found to be flanked by insertion sequences (IS5 family transposons). This suggests that the intervening DNA segment that contains the *porR* gene can be transposed and subjected to conjugative transfer. Thus, the seven genes might be co-transferred via horizontal gene transfer. Similar to the BI tree of PorR, the BI tree of UgdA does not exhibit monophyletic clades for all major classes under Bacteroidetes (both are a part of the seven genes). Both BI trees also exhibit similar topology where the four identified clusters with strong support have similar relative positions to each other in both BI trees. Other than the BI tree of PorR, the BI trees of the other 19 components also exhibit evidence of horizontal gene transfer. However, for the genes that encode those 19 components, they might undergo horizontal gene transfer less frequently compared to *porR* because the intervening DNA segment that contains *porR* is easily exchanged between bacteria under Bacteroidetes due to the presence of insertion sequences (IS5 family transposons) that flank it. This work also found other species under Chitinophagia, Saprospiria, and those that are unclassified that acquired T9SS component protein homologs that, to our knowledge, might not have been reported.

## ACKNOWLEDGEMENTS

We thank the Centre for Bioinformatics Research (CBR) for providing the facilities to conduct the bioinformatics analyses. We thank the reviewers for their comments on previous drafts of the manuscript.

### Funding

The authors received funding from the Universiti Kebangsaan Malaysia (UKM) grant, GUP-2018-147, which supported this research. The Titan V GPU was donated by NVIDIA Corporation. The funders had no role in study design, data collection and analysis, decision to publish, or preparation of the manuscript.

### Grant Disclosures

The following grant information was disclosed by the authors:
Universiti Kebangsaan Malaysia (UKM): GUP-2018-147.
Titan V GPU was donated by NVIDIA Corporation.

### Competing Interests

The authors declare there are no competing interests.

## Author Contributions

- Reeki Emrizal conceived and designed the experiments, performed the experiments, analyzed the data, prepared figures and/or tables, authored or reviewed drafts of the paper, and approved the final draft.
- Nor Azlan Nor Muhammad conceived and designed the experiments, authored or reviewed drafts of the paper, sourced funding for project, and approved the final draft.

## Data Availability

The raw data and sequence accessions are available in the Supplemental File.

## Supplemental Information

Supplemental information for this article can be found online at http://dx.doi.org/10.7717/peerj.9019#supplemental-information.

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
