# Peer review of "Phylogenetic comparison between Type IX Secretion System (T9SS) protein components suggests evidence of horizontal gene transfer"

_PeerJ, doi:10.7717/peerj.9019_

## Round 0.1 · original submission · Major Revisions

Dear Drs. Emrizal and Muhammad:

Thanks for submitting your manuscript to PeerJ. I have now received two independent reviews of your work, and as you will see, the reviewers raised some concerns about the research. Despite this, these reviewers are optimistic about your work and the potential impact it will have on research communities studying type IX secretion systems. Thus, I encourage you to revise your manuscript, accordingly, taking into account all of the concerns raised by both reviewers.

Please especially note that both reviewers struggled with several aspects of the presentation of your work, including clarity (undefined abbreviations, obscured information in figures, delivery of background, etc.) missing references, poorly-defined approach, and gross overstatements regarding the findings.

It seems that enough evidence was previously recorded on the lateral transfer of por genes given their limited distribution in Bacteroidetes. Your title and findings do not appear to be novel, so it would be better to not directly state this in your title or present the research in such a way that it appears you have discovered this novel finding.

Please perform more robust phylogenetic and comparative genomics analyses for your revision. Poor bootstrap support should be focused on (i.e., run jack-knifing, longer replications, alternative taxon sampling). Alternative optimality criteria (per reviewer 1’s suggestions) should also be employed. Please do more exhaustive searches of public databases for por genes and make it clear that a wide range of taxonomic groups were exhaustively evaluated.

I look forward to seeing your revision, and thanks again for submitting your work to PeerJ.

Good luck with your revision,

-joe

·

Basic reporting

1 - some parts of the manuscript are hard to understand. I think that one of the difficulties it to understand the area's terms. For instance, all the acronyms should be written out, even if you that they are obvious, such as IM, OM, SeqYeg, Wbp.

2 - no comment

3- Figures 2 - 5 have no sufficient resolution to see the bootstrap values or the names of the organisms. If the names are important, you should ensure that your reader can read them.

4 - no comment

Experimental design

1 - no comment

2 - The authors claim that the study "aims to investigate the phylogenetic relationship and taxonomic distribution between putative members of T9SS component protein families". Lines 46 - 47. I think that their problem was not limited to the inference of phylogenies. If I understood, they searched for evidence of horizontal transfer of T9SS. However, they already knew that such proteins were acquired horizontally from other bacteria and at the end, they concluded that only PorR was "acquired by T9SS-acquiring bacteria through horizontal gene transfer" (line 56).

3 - no comment

4 - Methods were described with sufficient detail. However, they built ML trees with only 100 bootstrap replications. Once they found several clades with low support, It would be hard to find the same topologies in a second round of each analysis. I suggest that they re-perform the phylogenies, but using a Bayesian approach with a minimum of 50 millions of Markov Chain steps. They can easily do it using MrBayes at the Cipres Gateway (https://www.phylo.org/), that saves time and provides very good results.

Validity of the findings

1- no comment

2- I disagree with the conclusion that 19 of the 20 genes provide trees where the OTUs were "clustered based on their classes under Bacteroidetes phylum". Hence, Figs. 2A - 2F, 3A-3C; 4A, 4C,4D ,4E; and 5A-5C present at least one OTU out of its expected cluster. If it is true that al T9SS were acquired from other bacteria, so the ancestral of each "monophyletic group of sequences" acquired the genes and passed them to their descendants. Therefore, I understand that such "outsiders" inherited the gene from bacteria of other groups, which should also be interpreted as a recent horizontal transfer.

_ Another point to be highlighted is that authors treated as monophyletic groups, clades with very low bootstrap support [e.g. group 3 of Fig. 2A (Bootstrap - BT = 65%); groups 1 and 2 of Figs. 2B and 2C (BT = 55 and 70%, respectively); group 3 of Fig. 3A (BT = 72%); group 4 of Fig. 3B (BT = 67%); group 5 of Fig. 3B (BT = 30%)]

3 - I think the authors focused only on PorR and should mention that most of the other proteins also show evidence of recent horizontal transfers (after the first transfer to Bacteroidetes or even the first transfer for each of its families). However, they showed evidence that PorR and its neighbors are flanked by IS5 transposons, which is very nice if it is a new finding. I think that the proper way to treat this point is that this operon was easily exchanged between the bacteria families (explaining why the figure 6 is so different from the other trees).
- I'm curious about why they did not test one of the neighbors, such as PorS, only to show that it has the same topology than PorR and reinforce that the recent horizontal transfer took place because of the IS5.

4 - no comment

Additional comments

Even though I have pointed out several flaws in the article, I know how hard is to mining genomes to search for protein families and define the homologous sequences to infer the phylogenies. Because of that, I recommended major revisions, once I would like to see a new version of this article.

I also would suggest that authors use a clearer way to explain their point. I'm not a specialist in cell membrane's proteins and found very difficult to follow some parts of the introduction. I think that a broad audience journal such as PeerJ has lots of readers that would be interested in recent horizontal gene transfer between pathogenic bacteria. Please, consider to re-wright the introduction to the sake of clarity.

Reviewer 2 ·

Basic reporting

In this paper, phylogenetic analysis revealed that PorR is different to other Por component proteins. This conclusion is not surprising. Because, PorR protein has similarity to the degT gene product of Bacillus stearothermophilus in Gram-positive bacteria and the homologs are present in broad (Shoji et al. 2002 Microbiology. 148(Pt 4):1183-1191.). In other words, PorR protein is not specific in Bacteroidetes phylum. On the other hand, PorT protein and other T9SS component proteins are specific in Bacteroidetes phylum. Therefore, the conclusions shown in this paper are not contrary to expectations.

Experimental design

No comment

Validity of the findings

In the last part of the results, authors found that additional species in Bacteroidetes phylum lack the proteins of PorG, PorL PorN, PorP, PorQ, PorT, PorU, and PorZ. This finding is interesting. However, it needs to be discussed with more experimental findings.

Additional comments

In addition, I present some advises to improve the manuscript.

1. As I mentioned above, PorR homologs are shown in Shoji et al. (2002) in Microbiology. The paper should be referred.

2. It is difficult for new readers to understand many PGN_ or PG_ numbers. Recently, T9SS related proteins are designated by Naito et al. (2019). [Naito M, Tominaga T, Shoji M, Nakayama K. PGN_0297 is an essential component of the type IX secretion system (T9SS) in Porphyromonas gingivalis: Tn-seq analysis for exhaustive identification of T9SS-related genes. Microbiol Immunol. 2019 63(1):11-20.] I recommend that gene numbers shown in the results are changed to PorF, SigP, PorE, and PorG as appropriate. [PG_0534, PGN_1437: PorF], [PG_0162, PGN_0274: SigP], [PG_1058, PGN_1296: PorE], [PG_0189, PGN_0297: PorG], and the paper of Naito et al. (2019) should be also referred.

3. In Figure 1, L3 are drawn by two rectangles. It should be drawn by three rectangles because small 3 in the right of L indicates the number of PorL.

4. In Figure 1, A-LPS is drawn outside of outer membrane. This seems to be weird. It should be drawn on the outer membrane.

5. In Figure 1, as the representative of T9SS cargo proteins, Kgp is shown. It should be changed to RgpB because RgpB is known to be clearly modified with A-LPS.

6. In Figure 1, “Lipid A transport pathway “should be changed to “LPS transport pathway”.

7. In Figure 1, R indicating PorR is colored in dark yellow. It should be changed to another different color. Because, the color is same to SigP. PorR function is known to be different to SigP one.

8. In Figure 8, it is drawn that PorR and PGN_0300 belong to regulation colored in red. PorR should be drawn as A-LPS synthesis shown by another color. PGN_0300 also should be drawn as unknown shown by color in grey or another color. PGN_0300 may be written as Omp17 or Skp (Taguchi et al. 2015).

9. In line 166 of the main text, “PorR might be acquired by T9SS-acquireing bacteria thorough horizontal gene transfer”. It seems overstatement. I recommend you change that “PorR might be acquired by other bacteria thorough horizontal gene transfer”.

10. In materials and methods section of the main text, ATCC 33277 should be non-italic font. Likewise, Rhodothermus marinus (italic font) DSM 4252 (non-italic font) and Salinibacter ruber (italic font) DSM 13855 (non-italic font) .

11. In line 353 of the main text, “porR gene” should be changed to “porR (italic font) gene”.

12. In line 364 of the main text, “similar to wzx “should be changed to “similar to wzx (italic font)“.

---

## Round 0.2 · Minor Revisions

Dear Drs. Emrizal and Muhammad:

Thanks for revising your manuscript. Reviewer 2 is satisfied with your revision; however, Reviewer 1 is unhappy with your phylogenetic analysis. Please address the concerns provided by Reviewer 1. Understand that we may enlist a third reviewer to ensure that your work is fairly reviewed. I believe that your attention to improving the phylogenetic analysis and your interpretations of this work will move your manuscript closer to acceptance for publication.

Please address these issues ASAP so we may move towards acceptance of your work.

Best,

-joe

·

Basic reporting

The authors had improved the article according to former recommendations. They inferred the phylogenetic trees using Bayesian Inference using appropriated models of amino acid substitution.

However, their findings were not completely convincing. They infer phylogenetic hypotheses of 20 T9SS proteins using Bayesian Inference (BI) and found different topologies (as expected). They argued that 19 of the 20 topologies presented "well-supported clusters" for the three major classes under Bacteroidetes. I have concerns about the concept of a cluster: "The definition of a cluster in this work is the cluster formed by majority of the terminal nodes of the tree (> 50%) that belong to the same class under Bacteroidetes" (lines 285 and 286). They used this particular concept instead of monophyletic clades containing the Bacteroidetes classes. The authors cited the phylogenetic inference of Karlsson et al. (2011) based on the 16S sequences as a reference for the phylogenetic relationships among Bacteroidetes. I consulted the referred article and could not find the three Bacteroidetes classes as monophyletic clades (it is not my research field, so I had difficulties to find the clusters). I think the authors could reproduce the 16S tree coloring the OTUs according to the cited classes.

Because of their particular concept of clusters, they found such "clusters" in 19 of the 20 trees but had to construct a table (Table S1), including all OTUs that grouped in unexpected clusters. Also, they compared the PorS and PorR trees and argued that they have the same "clusters" of Cytophagia (blue) and Bacterioidia (green). The problem with this state is that the "conserved clades" of PorS tree contain only one genus of Bacterioidia (Porphyromonas) and only one genus of Cytophagia (Hymenobacter).

Experimental design

The authors treated as "well suppported", clusters with posterior probability (PP) values > 0.8 (line 288), which is unusual since we consider as well supported clusters with PP > 0.95 when using BI.

In Abstract and Discussion, the authors refer to clusters in terms of their position at the printed figure, which is unusual in phylogenetic studies. At lines 48-49 they stated that: "General topology of PorR BI tree is different from the other 19 BI trees that are multifurcating at its centre into three big clusters that are approximately 60o apart from each other except for PorN BI tree that has two big clusters that are almost 180o apart. However, PorR BI tree has one big cluster at bottom half of the tree and branches sequentially diverging out from its centre at top half of the tree". At lines 397-399, they repeated the same idea: "This is because the 19 BI trees are multifurcating at its centre into three big clusters that are approximately 60o apart from each other (or two big clusters that are almost 180o apart from each other for BI tree of PorN) (Figs. 2-6). However, the BI tree of PorR has one big cluster at bottom half of the tree and branches sequentially diverging out from its centre at top half of the tree (Fig. 3)".

Validity of the findings

I could not understand the point of the horizontal transfer of genes in this work. It would be useful to inform (or infer) the consequences of the transfers for the organisms that received the genes. I think that authors had a great job in working with the Bayesian Inference. Still, I feel that the work would gain with the aid of a phylogenist to help them to describe the trees and interpret their results.

Additional comments

When I evaluated the first version of the manuscript, there were problems with the figure resolution as well as concerns about the methods used to infer the phylogenies and interpret the results. The authors resolved the first part of such problems, and now it is possible to see the results. However, I still have concerns about the interpretation of the results. My main suggestion is that the authors invite a phylogenist to help them interpret their findings. Also, it will be necessary to investigate the possible consequences of horizontal transfers of some of the T9SS family genes among Bacteroidetes.

Reviewer 2 ·

Basic reporting

I believe that the revised paper entitled as “Phylogenetic comparison between Type IX Secretion System (T9SS) protein components suggest evidence of horizontal gene transfer (#38397)’’ is very improved.
However, I recommend the authors to change one thing to improve as shown in general comments for the author.

Experimental design

no comment

Validity of the findings

no comment

Additional comments

I recommend the authors to change one thing to improve.
1. In Figure 1, A-LPS attachment property should be RgpB, not Kgp. This must be a mistake.

---

## Round 0.3 · Minor Revisions

Dear Drs. Emrizal and Muhammad:

Thanks for revising your manuscript. I am happy with your responses to the reviewers. However, a quick look at your manuscript indicates that the English is still in need of improvement. Do you have someone that can help? PeerJ has a service if this is something you would like to explore.

Please either have an English expert proof your manuscript or contact PeerJ about the service. Once this is done, we may move forward to accepting your work.

Best,

-joe

---

## Author Rebuttal · Round 0.3

**Centre for Bioinformatics Research**
Institute of Systems Biology (INBIOSIS)
Universiti Kebangsaan Malaysia
43600 UKM Bangi
Selangor Darul Ehsan, Malaysia
norazlannm@ukm.edu.my

February 13th, 2020

Dear Editors,

We thank the reviewers for their comments on the manuscript and we have edited the manuscript to address their concerns.

We hope that the manuscript is now suitable for publication in PeerJ.
* * *
Nor Azlan Nor Muhammad

On behalf of all authors.

**Reviewer 1 (Karla Yotoko)**

**Basic reporting**

*1) I have concerns about the concept of a cluster: "The definition of a cluster in this work is the cluster formed by majority of the terminal nodes of the tree (> 50%) that belong to the same class under Bacteroidetes" (lines 285 and 286). They used this particular concept instead of monophyletic clades containing the Bacteroidetes classes.*

In the previous manuscript, monophyletic clades concept is used in addition to that definition. In order to make it clearer, the manuscript has been revised; the definition is removed to avoid confusion and the monophyletic clade is added to replace it (Lines 290-294, 303-305, 367-368, 425-426, 475-476).

*2) I think the authors could reproduce the 16S tree coloring the OTUs according to the cited classes.*

The BI tree of 16S rRNA has been reproduced. The method (Lines 216-251), result (Lines 324-344), and discussion (Lines 400-424) sections of the manuscript have been updated to reflect that. Figure 7 is added to illustrate the constructed 16S rRNA tree.

*3) They compared the PorS and PorR trees and argued that they have the same "clusters" of Cytophagia (blue) and Bacterioidia (green). The problem with this state is that the "conserved clades" of PorS tree contain only one genus of Bacterioidia (Porphyromonas) and only one genus of Cytophagia (Hymenobacter).*

Due to limited number bacterial species were found to acquire PorS protein homologs in this study, we have decided to construct the BI tree for UgdA instead. Both *porR* and *ugdA* genes are within the intervening DNA segment that is flanked by insertion sequences. Both genes are involved in the Wbp pathway, which is one of the pathways in the A-LPS biosynthesis. The method (Lines 266-280), result (Lines 361-375), and discussion (Lines 470-487) sections of the manuscript have been updated to reflect that. Figure 9 is added to illustrate the comparison between PorR and UgdA BI trees.

**Experimental design**

*1) The authors treated as "well suppported", clusters with posterior probability (PP) values >
0.8 (line 288), which is unusual since we consider as well supported clusters with PP > 0.95
when using BI.*

In the current revision, only branches with PP > 0.95 are considered having a strong support.
Those branches are denoted as black coloured branches in Figs. 2-7 and Fig 9. In those figures,
the identified monophyletic clades that formed by terminal nodes that belong to the same class
under Bacteroidetes are denoted by solid curves. The monophyletic clades or its subclades with
a strong support are denoted by dashed curves. The reason behind those annotations is because
we would like to cater for the reader that is interested to see only the clades with a high support
and also for the reader that is interested to see the clades with/without a high support as there
is still a chance for those clades to appear even though the probability is low. The result (Lines
290-294, 330-333) section of the manuscript has been updated to reflect that.

*2) In Abstract and Discussion, the authors refer to clusters in terms of their position at the
printed figure, which is unusual in phylogenetic studies. At lines 48-49 they stated that:
"General topology of PorR BI tree is different from the other 19 BI trees that are multifurcating
at its centre into three big clusters that are approximately 60o apart from each other except
for PorN BI tree that has two big clusters that are almost 180o apart. However, PorR BI tree
has one big cluster at bottom half of the tree and branches sequentially diverging out from its
centre at top half of the tree". At lines 397-399, they repeated the same idea: "This is because
the 19 BI trees are multifurcating at its centre into three big clusters that are approximately
60o apart from each other (or two big clusters that are almost 180o apart from each other for
BI tree of PorN) (Figs. 2-6). However, the BI tree of PorR has one big cluster at bottom half
of the tree and branches sequentially diverging out from its centre at top half of the tree (Fig.
3)".*

Agreed. The instances of such statements in abstract and discussion sections are removed in
the revised manuscript.

**Validity of the findings**

*1) I could not understand the point of the horizontal transfer of genes in this work. It would be useful to inform (or infer) the consequences of the transfers for the organisms that received the genes. I think that authors had a great job in working with the Bayesian Inference. Still, I feel that the work would gain with the aid of a phylogenist to help them to describe the trees and interpret their results.*

We combine our response below.

**Comments for the authors**

*1) However, I still have concerns about the interpretation of the results. My main suggestion is that the authors invite a phylogenist to help them interpret their findings. Also, it will be necessary to investigate the possible consequences of horizontal transfers of some of the T9SS family genes among Bacteroidetes.*

The aim of the work is to report about the phylogeny of T9SS protein components that hasn't been covered yet by the literature as most works on this bacterial secretion system have been focusing on functional characterisation of its protein components. We observed that the BI tree of PorR is rather different from the other 19 BI trees of T9SS protein components. Thus, this work also aims to establish the differences between the BI of PorR tree and other 19 BI trees. The point of the horizontal gene transfer in this work is just a mean to explain the differences that we observed between the BI tree of PorR and other 19 BI trees. It is interesting to infer the possible consequences of such transfer but we afraid that we don't know any close expert that we could invite to interpret the result from that perspective.

**Reviewer 2**

**Comments for the authors**

*1) I recommend the authors to change one thing to improve. In Figure 1, A-LPS attachment property should be RgpB, not Kgp. This must be a mistake.*

Figure 1 has been edited to reflect that (Figure 1).

---

## Round 0.4 · Minor Revisions

Dear Drs. Emrizal and Muhammad:

Thanks for once again revising your manuscript. However, I did not see your reply to the comments of Reviewer 1. Also, is the concern of Reviewer 2 fixed?

Please provide your rebuttal letter so we can ensure that you addressed the concerns of the reviewers and move forward with accepting your work for publication.

Thanks,

-joe

---

## Round 0.5 · accepted · Accept

Dear Drs. Emrizal and Muhammad:

Thanks for re-submitting your revised manuscript to PeerJ, and for addressing the English issues. I now believe that your manuscript is suitable for publication. Congratulations! I look forward to seeing this work in print, and I anticipate it being an important resource for research communities studying type IX secretion systems.

Thanks again for choosing PeerJ to publish such important work.

-joe

---

## Author Rebuttal · Round 0.5

**Centre for Bioinformatics Research**
Institute of Systems Biology (INBIOSIS)
Universiti Kebangsaan Malaysia
43600 UKM Bangi
Selangor Darul Ehsan, Malaysia
norazlannm@ukm.edu.my                                    March 20th, 2020

Dear Editor,

We have edited and had the manuscript proofread to address your concerns.

We hope that the manuscript is now suitable for publication in PeerJ.
* * *
Nor Azlan Nor Muhammad

On behalf of all authors.

# *Certificate of Acknowledgement*

We acknowledge that

## Dr Nor Azlan Nor Muhammad

used our Proofreading service recently for the 21.5 page(s) of Journal Article with the title as follows:

## *PHYLOGENETIC COMPARISON BETWEEN TYPE IX SECRETION SYSTEM (T9SS) PROTEIN COMPONENTS SUGGESTS EVIDENCE OF HORIZONTAL GENE TRANSFER*

On

## 19 March 2020

Acknowledgement by:

**MUHAMMAD ZAKI RAMLI**
**FOUNDER OF PROOFREADERS UNITED**